# Advantages of Adult Mouse Dorsal Root Ganglia Explant Culture in Investigating Myelination in an Inherited Neuropathic Mice Model

**DOI:** 10.3390/mps5040066

**Published:** 2022-07-22

**Authors:** Yun Jeong Mo, Yu-Seon Kim, Minseok S. Kim, Yun-Il Lee

**Affiliations:** 1Well Aging Research Center, Division of Biotechnology, Daegu Gyeongbuk Institute of Science and Technology (DGIST), Daegu 42988, Korea; myj214@dgist.ac.kr (Y.J.M.); yuseon25@dgist.ac.kr (Y.-S.K.); 2Department of New Biology, Daegu Gyeongbuk Institute of Science and Technology (DGIST), Daegu 42988, Korea; kms@dgist.ac.kr

**Keywords:** dorsal root ganglia explant, Charcot-Marie-Tooth, myelination, co-culture

## Abstract

A co-culture of neurons and Schwann cells has frequently been used to investigate myelin sheath formation. However, this approach is restricted to myelin-related diseases of the peripheral nervous system. This study introduces and compares an ex vivo model of adult-mouse-derived dorsal root ganglia (DRG) explant, with an in vitro co-culture of dissociated neurons from mouse embryo DRG and Schwann cells from a mouse sciatic nerve. The 2D co-culture has disadvantages of different mouse isolation for neurons and Schwann cells, animal number, culture duration, and the identification of disease model. However, 3D DRG explant neurons and myelination cells in Matrigel-coated culture are obtained from the same mouse, the culture period is shorter than that of 2D co-culture, and fewer animals are needed. In addition, it has simpler and shorter experimental steps than 2D co-culture. This culture system may prove advantageous in studies of biological functions and pathophysiological mechanisms of disease models, since it can reflect disease characteristics as traditional co-culture does. Therefore, it is suggested that a DRG explant culture is a scientifically, ethically, and economically more practical option than a co-culture system for studying myelin dynamics, myelin sheath formation, and demyelinating disease.

## 1. Introduction

The myelin sheath provides a multi-insulating and protective layer that covers the axons of neurons to promote the transmission of electrical impulses and provide nutrients. It is generated by Schwann cells in the peripheral nervous system (PNS) from oligodendrocytes in the central nervous system.

To investigate the processes of myelin formation and demyelination in the PNS, several in vitro culture models have been developed. Most researchers use a co-culture system of dissociating neurons from mouse embryo dorsal root ganglia (DRG) and mouse sciatic nerve Schwann cells to study the biological, physiological, and pathological mechanisms of myelination and demyelinating disease [1,2]. However, the co-culture model system can introduce mixed genetic backgrounds (knock-out, natural mutants, and wild-type (WT)) to the study of molecular and cellular events and mechanisms that involve inherited demyelinating disorders, because each cell type is derived from different mice [3,4]. Previously, studies of peripheral nerve myelination have used a mouse embryonic DRG explant culture model that contains sensory neurons and satellite glial cells to exclude mixed genotypes and phenotypes [5,6]. Transcriptome analysis showed that satellite glial cells from DRG and sciatic-nerve-derived Schwann cells were notably similar [7]. An organotypic ex vivo model of adult-mouse-derived DRG reportedly explains neuroplasticity, neuronal–glial interaction, and neuritogenesis [8].

In the present study, we describe the procedures for producing adult mouse DRG explant cultures, including neurons and satellite glial cells, with a Matrigel coating and 3D-like encapsulation for regeneration as an extracellular microenvironment and compare with a co-culture with dissociated DRG neurons from day 13 mouse embryos and purified Schwann cells from adult mouse sciatic nerves (Figure 1A,B) [9]. The results apply to an investigation of myelin sheath formation in PNS demyelinating diseases, such as Charcot–Marie–Tooth (CMT) disease, a slow but progressive genetic disorder. We also discuss the advantages of adult DRG explant culture as a tool for investigating the disease mechanism of demyelination in a mouse model.

CMT disease remains incurable, despite considerable research. An important experimental therapeutic strategy requires optimal cell models to study both the treatments and mechanisms of demyelination. Many researchers developed in vitro CMT cell models for effective screening. Several precedents, such as the CMT2E model, which overcomes the limitations of 2D construction by applying a 3D spheroid method [10], the CMT4B2 model using embryos of mutant mice [11], and the murine therapeutic model for CMT1 and CMT4, exist [12].

This research focused on the design of an in vitro CMT cell model that can be applied in multiple ways for screening/mechanism studies. One CMT mouse model, Trembler-J (Tr-J) from Jackson Laboratory (strain no. 002504), involves spontaneous mutants carrying point mutations in the *PMP22* gene [13]. The model is only capable of identifying mutations through normal genotyping and offering different degrees of severity, depending on the homo/heterozygous carrier. After a month or two, a standard disease model was found using the phenotype and behavior of mice.

Therefore, an adult mouse DRG explant culture, including neurons and satellite glial cells, is a more effective technique for studying myelin dynamics, myelination, and the mechanism of demyelination. This experimental cell model system for CMT mice may greatly enhance the production of potential drug candidates through high-throughput screening.

## 2. Experimental Design

This technique describes how to explant culture adult DRG for myelination induction. Adult DRG was cultured from 3- to 5-month-old WT C57BL/6J and Tr-J CMT disease model mice to investigate myelin sheath formation and the mechanism responsible for the demyelinating disease. Both male and female mice were used to study myelination and demyelination. Isolated lumbar DRGs were transferred to the center of the Matrigel-coated culture dish and induced myelination by following the steps.

### 2.1. Materials

Neurobasal (Fisher Scientific, Waltham, MA, USA; Cat. no. 21103-049)MEM (Fisher Scientific, Waltham, MA, USA; Cat. no. 11095-080)DMEM/F12 (Welgene, Gyeongsan, Korea; Cat. no. LM 002-04)DMEM (Welgene, Gyeongsan, Korea; Cat. no. LM 001-05)B27 supplement (Fisher Scientific, Waltham, MA, USA; Cat. no. 17504-044)N_2_ supplement (Fisher Scientific, Waltham, MA, USA; Cat. no. 17502-048)Bovine pituitary extract (Fisher Scientific, Waltham, MA, USA; Cat. no. 13028-014)l-Glutamine (Fisher Scientific, Waltham, MA, USA; Cat. no. 25030-081)d-Glucose (Millipore-Sigma, Burlington, MA, USA; Cat. no. G7528)l-Ascorbic acid (Millipore-Sigma, Burlington, MA, USA; Cat. no. A5960)NGF (Alomone Labs, Jerusalem, Israel; Cat. no. N-100)Horse serum (Fisher Scientific, Waltham, MA, USA; Cat. no. 26050-088)FBS (PEAK, Wellington, CO, USA; Cat. no. PS-FB1)Antibiotic–Antimycotic (A/A) (Welgene, Gyeongsan, Korea; Cat. no. LS 203-01)DPBS (Welgene, Gyeongsan, Korea; Cat. no. LB 001-02)Matrigel (BD, Franklin Lakes, NJ, USA; Cat. no. 356231)

### 2.2. Equipment

Fine forceps (Ideal-tek, Chiasso, Switzerland; Cat. no. 5A.SA.0)Spring scissor (Jeungdo, Seoul, Korea; Cat. no. S-1135)12-well plate (SPL, Pocheon, Korea; Cat. no. 30012)60 mm dish (SPL, Pocheon, Korea; Cat. no. 10060)37 °C CO_2_ incubator (Fisher Scientific, Waltham, MA, USA; Cat. no. HERAcell 240i)Clean bench (LABCONCO, Kansas, MO, USA; Cat. no. 6’ Purifier Logic+ Class 2 A2)Axio observer Z1 (Carl Zeiss, Oberkochen, Germany; Cat. No. Observer.Z1)

## 3. Procedure

### 3.1. Dish Preparation for DRG Explant Culture

Cooling to −20 °C, a tip, tube, dish, and plate.
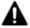
**CRITICAL STEP** This step is essential to keep the Matrigel in liquid form.Matrigel and ice-cold DMEM were mixed in a ratio of 1:3 and incubated at 4 °C in liquid form to be coated on surface of the plate [14].As such, 1 mL of Matrigel and DMEM solution was applied to one well (on a 12-well plate).After completely covering the plate, 950 µL of 1 mL of the mixed solution was rapidly retrieved at 4 °C and reused to coat the other plates. For the experimental procedure, the amount of solution needed was modified according to the plate surface area.The remaining 50 µL of Matrigel solution was incubated at 37 °C for at least 1 h (overnight is recommended).
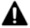
**CRITICAL STEP** If the gelation time exceeds 36 h, the Matrigel coating can be too dry to use.

### 3.2. Adult DRG Isolation

A 3–5-month-old male/female mouse is used. After euthanasia by cervical dislocation, the vertebral column was surgically removed from each mouse body.
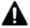
 CRITICAL STEP The last rib was marked to classify DRG number, the upper part of the pelvis was included to isolate all lumbar DRGs (L DRG), and the vertebrae from the last rib were counted to identify the L DRG.The spinal column was dissected with a sharp blade at the vertebral foramen, following the spinous process, to cut through the center of the vertebrae.The two separated sections were transferred to a Petri dish with sterile 1× DPBS. The spinal cord was gently removed with fine forceps to view the nerve fiber connected to the DRG.The inner portion of the vertebral foramen was exposed after dislodging the spinal cord with a pair of forceps. The forceps’ tip was scratched to disconnect the membrane tissue. The bilateral lumbar DRG (L1–L6) was then collected with micro-spring scissors.
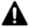
 CRITICAL STEP The location of the last thoracic vertebrae (=13 thoracic vertebrae, T13 vertebrae) was identified from the last rib. As seen in Figure 1A, the last rib was located on the cranial T13 vertebrae and the T13 DRG was placed on the caudal T13 vertebrae.Isolated lumbar DRGs were transferred directly to a 100 mm Petri dish containing 1× DPBS with fine forceps. The nerve fibers were removed with spring scissors from the connecting DRG to improve the purity of the culture.
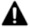
 CRITICAL STEP The capsule membrane tissue around the DRG was peeled off.

### 3.3. Adult DRG Explant

Before the DRG was placed on the plate, a Matrigel-coated well was pre-treated with 200 μL of neuron growth (NG) media to prevent drying out.
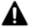
**CRITICAL STEP** The culture dish was shaken gently to ensure that the Matrigel coating was not damaged or dislodged.The DRG was transferred to the center of the Matrigel-coated culture dish with fine forceps. The isolated DRG was then carefully and slightly buried in the Matrigel to protect against DRG detachment and incubated for 1 h in a CO_2_ incubator at 37 °C. The CO_2_ level was maintained at 5% in all culture steps.After incubation, 300 µL of NG media was added for a total volume of 500 µL.
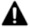
**CRITICAL STEP** The placed DRG was completely attached under NG media. The dry DRG was dislodged at the explanted point and floated.To change the medium, 300 µL of cultured media was carefully removed with a 1 mL micropipette and 300 µL of new media was added slowly once every 3 days.
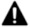
**CRITICAL STEP** Vacuum aspirator was not used to replace the NG media.

### 3.4. Media Conditions and Duration to Induce Myelination

The adult DRG was incubated with conditional media (Table 1) at each step (cell growth, differentiation, or myelination) for at least 35 days.
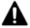
**CRITICAL STEP** When changing the step (e.g., cell growth to differentiation, differentiation to myelination), the previous step medium must be removed.
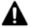
**CRITICAL STEP** A fresh stock of l-ascorbic acid was prepared at 2-week intervals. Because the l-ascorbic acid stock is rapidly oxidized, the lights on the clean bench were turned off when the myelination medium was changed.
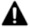
**CRITICAL STEP** When ascorbic acid was activated, part of the media turned from pink to yellow based on the media pH-indicating system (yellow indicating acidic pH) as the ascorbic acid stock was added to the myelination media. The color transition was checked to confirm the activity of ascorbic acid before changing the medium.
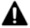
**CRITICAL STEP** The maximum stable period of forskolin stock is 3 months. As a final step, the myelin sheath formation was reduced if forskolin activity declined or became unstable. Adequate care was taken during the freezing storage period, to not extend the stable period of the forskolin stock.

### 3.5. Immunodetection to Observe the Myelin Sheath

Cultured adult DRG explants were rinsed twice in 1× PBS and fixed for immune analysis in the culture dish in ice-cold 4% paraformaldehyde solution in 1× PBS at room temperature (25 °C) for 20 min to observe the myelin sheath formation.To avoid dislodging the DRG explant, it was gently washed in 1× PBS for 3 min using a pipette. This step was repeated thrice.The adult DRG culture was incubated in a primary blocking solution at room temperature for 1 h and added to an antibody (Tuj1, myelin basic protein (MBP), contactin-associated protein 1 (Caspr)) in a primary antibody solution at 4 °C overnight (Table 2 and Table 3).At the end of the primary antibody incubation, it was washed three times for 5 min with 1× PBS.A secondary blocking was conducted at room temperature for 30 min.A secondary antibody solution was applied to the adult DRG explant culture at room temperature for 2 h (Table 2 and Table 3).
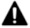
**CRITICAL STEP** This step should be protected from light.After secondary antibody incubation, it was removed with three 5 min 1× PBS.DAPI staining (1.25 μg in 1 mL PBS) occurred at room temperature for 10 min.The staining solution was removed with two 5 min washes of 1× PBS and 500 μL of 1× PBS was added before imaging the adult DRG explant culture with a Zeiss Live Cell system and ZEN software.

## 4. Expected Results

### 4.1. Several Coating Methods for Adult DRG Explant Culture

Poly-d-lysine (PDL)-laminin coating is frequently used for co-cultures [15], with 25% Matrigel coating for adult DRG explant culture. Additionally, a drop-based encapsulation method with 25% Matrigel was used. Adult DRG explants grown on the PDL-laminin coating (Figure 2A) on the 25% Matrigel coating (Figure 2B) and in the capsule of 25% Matrigel (Figure 2C) were cultured and compared with one another. Caspr is a marker of the paranodal region that appears at both ends of a mature myelin sheath, indicating mature myelination. In the myelin sheath formation, different coating types have different strengths and weaknesses (Table 4), but all three types were observed in mature myelination (Figure 2, lower-panel image). Additionally, both Matrigel coating types induced myelination compared with the PDL-laminin type. As shown in Table 4, the suitable coating types were selected according to the purpose of the experiment.

### 4.2. Expression of Schwann Cell Characteristics in Adult DRG Explant Culture

In a previous study, it was confirmed that myelination could be induced in adult DRG explant cultures. The researchers could also select suitable coating types for explant cultures and form mature myelin sheaths on any coating type. Next, adult DRGs from the Tr-J (CMT mouse model) were applied to the explant culture based on the fact that these experiments were used in an adult mouse. In immunocytochemistry results of myelination induced at co-culture with adult Tr-J Schwann cells and embryonic DRG neuron cells, MBP signal (green) forms appeared abnormal compared with those from WT mice culture (Figure 3C,D). The MBP signal from WT had a straight shape and uniform thickness without a break (Figure 3A,B,G,H), whereas that of Tr-J is a crumpled, broken form, with uneven thickness (Figure 3C,D,I,J). Under the same growth conditions, mature myelination was lower on Tr-J DRG explant culture (Figure 3I,J) than on WT DRG (Figure 3G,H). Additionally, Tr-J co-culture (Figure 3C,D) and Tr-J DRG explant culture (Figure 3I,J) were similar to a morphologically abnormal formation, which is a crumpled or broken form with uneven thickness.

In addition to the crumpled or broken form with uneven thickness in the Tr-J myelin sheath seen on immunocytochemistry images, the total length of the myelin sheath is shorter than that of WT. There is also a clear difference in the average lengths in Figure 3E,K, and the distribution levels are shown in graphs; Figure 3F,L are also markedly different. These results mean that the characteristics of the Schwann cell in co-culture and the satellite glia cell in adult DRG explant culture are similar. Conclusively, the adult DRG explant culture system with several advantages might be more effective to investigate the disease characteristics of the CMT mouse model.

## 5. Discussion

A method for adult DRG explant culture for mechanism study and screening of the CMT1E model is described in this research. There are several differences compared with traditional co-culture. The main difference is that both the neurons and Schwann cells originate from one individual mouse. Therefore, the experiments are relatively straightforward and fewer mice are used than in co-culture. The adult DRG explant system can, most likely, be applied to another neuropathy disease if it is caused by a genetic problem in Schwann cells, even if not CMT1E, because Schwann cells of co-culture and satellite glial cells of adult DRG explant culture can both reflect disease characteristics in vitro.

According to the previous results, the adult DRG explant culture is a powerful and effective alternative method for studying myelination, demyelination, neuronal–glial dynamics, and the pathophysiological mechanisms of PNS disease. An adult DRG explant culture may serve as a model system to prevent the possibility of mixed genetic backgrounds present in co-culture systems.

However, the adult DRG explant culture system has some limitations. First, it is uncertain and difficult to control the total number of cells. Additionally, the co-culture model clearly has only two types of cells, whereas the adult DRG explant model has more types of cells mixed. This may be an advantage in that it is similar to the in vivo environment, but simultaneously, it is a disadvantage because it is difficult to clearly identify the background.

The adult DRG explant culture system can still be an efficient model for investigating myelination in vitro, despite these limitations. This adult DRG explant culture might be a beneficial model system to investigate the potential therapeutic candidates for lack of myelination and inherited demyelinating disease.

## Figures and Tables

**Figure 1 mps-05-00066-f001:**
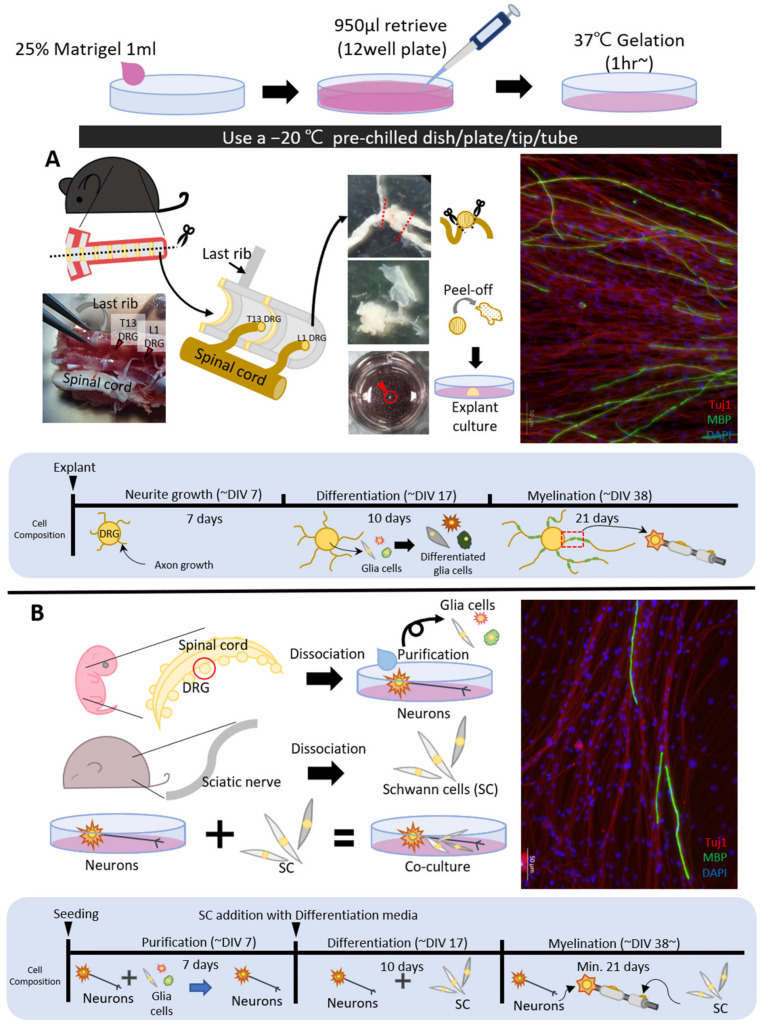
Comparison of co-culture and adult dorsal root ganglia (DRG) explant culture. The myelin basic protein (MBP) (green) signal appears as an immunocytochemistry result and means the myelin sheath. Blue, DAPI staining for DNA; red, Tuj1 staining for axons by immunofluorescence. Both adult DRG explant culture (**A**) and co-culture (**B**) show equal MBP signals on immunocytochemistry. Therefore, adult DRG explant culture can reduce the number of primary culturing experimental steps to one time and obtain the same results as co-culture. Scale bars = 50 μm.

**Figure 2 mps-05-00066-f002:**
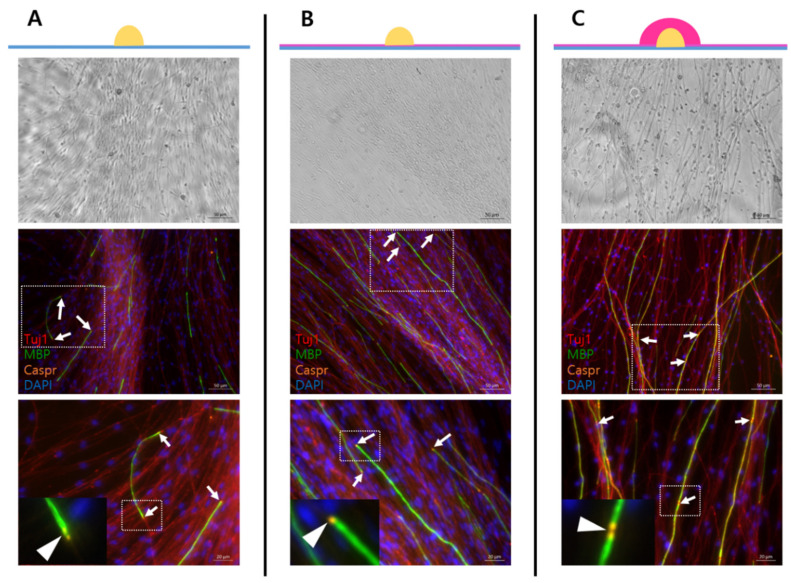
Different coating methods for the adult DRG explant culture. Blue, DAPI staining for DNA; red, Tuj1 staining for axon; green, MBP staining for myelin sheath; orange, Caspr staining for paranodal region, which is a marker of mature myelination. Myelination induced result at adult DRG explant culture on PDL-laminin coating dish (**A**), DRG explant culture on 25% Matrigel coating dish (**B**), and DRG encapsulated with 25% Matrigel (**C**). The upper-panel images were taken with bright-field microscopy. Middle-panel images resulted from immunocytochemistry; lower-panel images are more magnified than middle-panel images (white box). Upper and middle images scale bars = 50 μm, lower images scale bars = 20 μm. White arrows and arrowhead indicate a paranodal region (=Caspr).

**Figure 3 mps-05-00066-f003:**
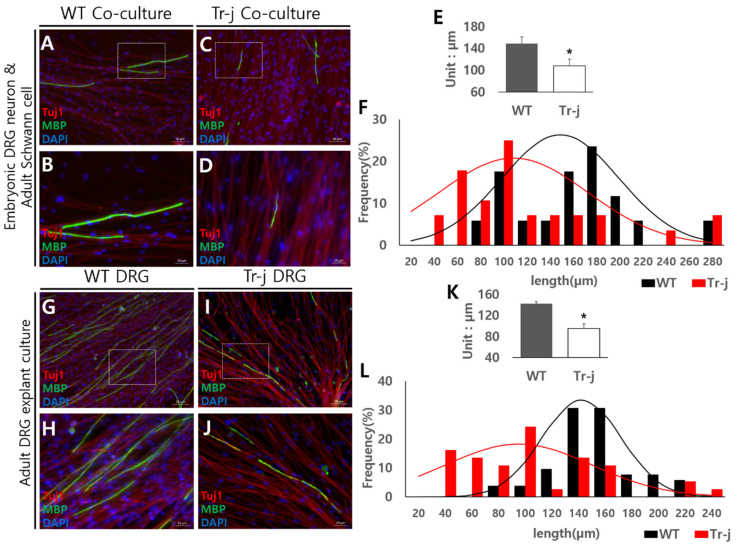
Expression of disease model characteristics in adult DRG culture. Blue, DAPI staining for DNA; red, Tuj1 staining for axon; green, MBP staining for myelin sheath. Myelination induced from Tr-J mouse cells. Co-culture with embryonic DRG neurons and adult wild-type (WT) Schwann cells (**A**,**B**). Co-culture with embryonic DRG neurons and adult Tr-J Schwann cells (**C**,**D**). Higher-magnification images of the white box portion of A and C (**B**,**D**). Myelin sheath length in a co-culture system, *: *p* values are less than 0.05 (**E**). Myelin sheath length distribution levels in co-culture (**F**). Adult WT DRG explant cultures (**G**,**H**). Adult Tr-J DRG explant culture (**I**,**J**). Higher-magnification images of the white box portion of G and I (**H**,**J**). Myelin sheath length in an adult DRG explant culture system, *: *p* values are less than 0.05 (**K**). Myelin sheath length distribution levels in an adult DRG explant culture system (**L**). The myelin sheath length appears shorter in Tr-J, both culture systems. (**A**,**C**,**G**,**H**) images scale bars = 50 μm. (**B**,**D**,**H**,**J**) images scale bars = 20 μm.

**Table 1 mps-05-00066-t001:** Culture step and media composition.

Step	Media Name	Duration	Composition
Cell growth	Neuron growth media (NG media)	7 days	Neurobasal Antibiotic–Antimycotic 1% d-Glucose 4 mg/mL B27 supplement 1× NGF 50 ng/mL l-Glutamine 2 mM
Differentiation	Differentiation media	7–10 days	DMEM/F-12 Antibiotic–Antimycotic 1% NGF 50 ng/mL N_2_ supplement 1× l-Glutamine 2 mM
Myelination	Myelination media	~21 days	MEM (Horse Serum 5%) Antibiotic–Antimycotic 1% d-Glucose 4 mg/mL Bovine pituitary extract 50 μg/mL Forskolin 0.5 μM NGF 50 ng/mL N_2_ supplement 1× l-Ascorbic acid 50 μg/mL l-Glutamine 2 mM

**Table 2 mps-05-00066-t002:** Product information for immunodetection.

	Antibody	Cat. No.	Titer
Primary	Tuj1	SYSY, 302–304	1:500
	MBP	Millipore, MAB386	1:200
	Caspr	Abcam, ab34151	1:500
Secondary	Cy5-Guinea pig	Jackson, 706-175-178	1:1000
	Cy3-Rat	Jackson, 712-165-150	
	Alexa 488-Rabbit	Jackson, 711-545-152	
	**Product name**	**Cat. No.**	**Final concentration**
DNA imaging	DAPI	Thermo, 62248	1.25 μg/mL

**Table 3 mps-05-00066-t003:** Buffer composition and recipe for immunodetection.

Solution	Recipe		Final Volume	Step
Primary blocking	10× PBS Donkey serum 10% Triton X-100 BSAD.W	1 mL 1 mL 300 μL 10 mg 7.7 mL	10 mL	3. 4. 3 Room temperature for 1 h
Primary antibody	10× PBS Donkey serum 10% Triton X-100 BSA D.W	1 mL 100 μL 300 μL 10 mg 8.6 mL	10 mL	3. 4. 3 Primary antibody dilution 4 °C overnight
Secondary blocking	10× PBS Donkey serum BSA D.W	1 mL 100 μL 10 mg 8.9 mL	10 mL	3. 4. 5 Room temperature for 30 min
Secondary antibody	10× PBS Donkey serum 10% Triton X-100 BSA D.W	1 mL 100 μL 100 μL 10 mg 8.8 mL	10 mL	3. 4. 6 Secondary antibody dilution Room temperature for 2 h

**Table 4 mps-05-00066-t004:** Strength and weakness according to coating type.

Coating Type	Strength	Weakness
PDL-laminin	Suitable for observation using a microscope	Difficult to attach on dish base when explant culture begins
25% Matrigel	Attachment to dish base is more accessible than that with PDL-laminin coating Matrigel degradation issue is not a serious issue compared to encapsulation with 25% Matrigel	It is difficult to clearly focus on when using a microscope because of different heights of cells (cells will burrow into Matrigel coating)
Encapsulated with 25% Matrigel	Easy to attach on dish base when explant culture beginning Fast growing	Not suitable for long-term culture (Matrigel degradation), disadvantage for observation using a microscope

## Data Availability

Not applicable.

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
