# Peer review of "Advantages of Adult Mouse Dorsal Root Ganglia Explant Culture in Investigating Myelination in an Inherited Neuropathic Mice Model"

_mps, 2022, doi:10.3390/mps5040066_

Round 1

Reviewer 1 Report

In this paper Mo et al., describe the development of a new DRG explant 3D Matrigel-coated system for the testing of new drugs for demyelinating neuropathies. The methodology presented is very interesting and can be easily replicated compared to Schwann cell co-cultures giving the opportunity for further investigation of demyelinating peripheral neuropathies.

The authors should include the paper of Maciel et al., 2019 Clinical Pharmacology and Therapeutics which refers to the use of a 2D cell culture system for the investigation of peripheral neuropathies and mainly for CMT2E.

Further, in Figure 3 authors must clarify in the figure legend the magnification of each picture. Also in the WT culture some small breaks are evident too similar to the CMT cultures.

Finally, discussion could be improved.

Author Response

Reviewer #1

In this paper Mo et al., describe the development of a new DRG explant 3D Matrigel-coated system for the testing of new drugs for demyelinating neuropathies. The methodology presented is very interesting and can be easily replicated compared to Schwann cell co-cultures giving the opportunity for further investigation of demyelinating peripheral neuropathies.

Response: We thank the reviewer for these comments. We have edited the text as the reviewer suggested.

The authors should include the paper of Maciel et al., 2019 Clinical Pharmacology and Therapeutics which refers to the use of a 2D cell culture system for the investigation of peripheral neuropathies and mainly for CMT2E.

Response: We have added this paper in the introduction and reference part as the reviewer commented.

Line 56. “Several precedents, such as the CMT2E model, which overcomes the limitations of 2D construction by applying a 3D spheroid method

Line 309. “10. Maciel R.; Correa R.; Bosso Taniguchi J.; Prufer Araujo I.; Saporta MA. Human Tridimensional Neuronal Cultures for Phenotypic Drug Screening in Inherited Peripheral Neuropathies. Clin Pharmacol Ther. 2020, 107(5), 1231-1239

Further, in Figure 3 authors must clarify in the figure legend the magnification of each picture. Also in the WT culture some small breaks are evident too similar to the CMT cultures. Finally, discussion could be improved.

Response: We have edited the figure legend as the reviewer commented.

Line 242~3 and 245~6. “Higher magnification images of the white box portion of A and C (B and D) ~ Higher magnification images of the white box portion of G and I (H and J).”

There are some limitations in checking the myelination status using immunocytochemistry in a culture system for myelination. Nevertheless, it can be distinguished from demyelinated animal models by myelin distribution, length of myelin, the node of Ranvier formation, etc. Thus it is being used for various purposes, such as developing therapeutic treatments and studying mechanisms. To clarify myelin sheath formation in WT vs CMT model culture, we have added a quantitative analysis in figure 3 and described these results in the manuscript. And we have amended the discussion section.

Line 231. “In addition to the crumpled or broken form with uneven thickness of the Tr-J myelin sheath seen on immunocytochemistry images, the total length of the myelin sheath is shorter than that of WT. ~~~ to investigate the disease characteristics of the CMT mouse model.

Reference

  1. A Intisar et al. (2022) Biofabrication 14(1)
  2. S Rangaraju et al. (2010) J Neuroscience 30 (34) 11388-11397

Reviewer 2 Report

The authors aim to show new model of DRG explant culture for studying myelin dynamics, myelin sheath formation, and demyelinating disease in comparison with a co-culture system. The protocol describes how to explant culture of the adult DRG for inducing myelination. The methodology is clear and concise and provide necessary information to repeat the method as well as critical steps. The figures outline the key message. However, limitation of the approach should be described.

Author Response

Reviewer #2

The authors aim to show new model of DRG explant culture for studying myelin dynamics, myelin sheath formation, and demyelinating disease in comparison with a co-culture system. The protocol describes how to explant culture of the adult DRG for inducing myelination. The methodology is clear and concise and provide necessary information to repeat the method as well as critical steps. The figures outline the key message. However, limitation of the approach should be described.

Response: We thank the reviewer for these positive comments. We constructively have addressed the limitations of this approach in the discussion part.

262 Line. “However, the adult DRG explant culture system has some limitations. First, it is uncertain and difficult to control the total number of cells. Additionally, the co-culture model clearly has only two types of cells, whereas the adult DRG explant model has more types of cells mixed. This may be an advantage in that it is similar to the in vivo environment, but simultaneously, it is a disadvantage because it is difficult to clearly identify the background.

Reviewer 3 Report

Mo and colleague provided a protocol on adult mouse DRG explant culture containing detail step by step protocol and provided some nice images. Although there are some 3D cocultures with DRG explants protocols already available, many uses embryonic tissue or uses novel biomaterials that is not readily available to other researchers. Also, this protocol seems to be quicker according to the authors.

There are major concerns that require addressing.

1)      The demyelinating disease in the title does not agree with the model provided since the model shows lack of myelination rather demyelination. Therefore, the model needs to show that there is myelination, then after an intervention, there is demyelination. Even better still, if give a treatment and the demyelination stops and myelination/remyelination occurs.

2)      The term ‘satellite Schwann cells’ (e.g. line 20, 40) used throughout the manuscript is incorrect as satellite cells are not Schwann cells. Furthermore, in lines 41-42, the authors wrote: ‘there is characteristically no difference between satellite Schwann cells of DRG and sciatic-derived Schwann cells’. This is incorrect as the paper (George et al., 2018) says: ‘similar rather than no difference’. Also, is it possible for the myelination to be due to the Schwann cells that have not been completely removed when the peripheral nerves were removed as the nerve fibers are present within the DRG?

3)      There are many grammatical errors throughout the manuscript making some parts confusing, so require editing by a native English-speaking scientist. For example, line 164: ‘declined become unstable’ and line 165 ‘not to be a stable period for forskolin stock’, and many others.

4)      Missing the strength of the protocol which I assume is a simpler protocol with just one step rather than multiple steps. Maybe this should be in the abstract or introduction.

5)      Removing the sheath surrounding the DRG is difficult, so what suggestion do the authors have as there are many methods?

6)      Line 170. How was ‘gently washed’ carried out? Pipette slowly? Any swirling or on a slow rocker? How was PBS removed between washes?

7)      (Table 2) organisation unclear and need to be split up into antibody table and reagent preparation protocol table. Better transpose the table so similar to table 1 format and have the volume in a separate column for neatness. Also, avoid abbreviation GP and Rb.

8)      Figure 2.  Use a few arrows to show cell body if possible to help reader see cells. Upper scale bars should be black as unclear in white.

9)      Figure 3. Unclear which part of the culture is adult or embryonic, and which is WT or Trembler J mouse? Also panel lettering ordering need to change (e.g. A and D should be A and B) for better flow in body text.

10)  Missing a discussion section to compare your methods with others.

Minor concerns:

11)  what’s the percentage of CO2 levels?

12)  Fig1. Last section. Should it be ‘neurite growth’ rather than ‘cell growth’?

13)  Define ICC.

14)  Line 115. Confused why 36h if only overnight is sufficient which is 24h.

15)  Line 150. ‘span’?

16)  Table 1. Define ‘A/A’

Author Response

Reviewer #3

Mo and colleague provided a protocol on adult mouse DRG explant culture containing detail step by step protocol and provided some nice images. Although there are some 3D cocultures with DRG explants protocols already available, many uses embryonic tissue or uses novel biomaterials that is not readily available to other researchers. Also, this protocol seems to be quicker according to the authors.

Response: We thank the reviewer for the careful reading of our manuscript and insightful suggestions. As the reviewer commented, we have amended the manuscript and the figures.

1) The demyelinating disease in the title does not agree with the model provided since the model shows lack of myelination rather demyelination. Therefore, the model needs to show that there is myelination, then after an intervention, there is demyelination. Even better still, if give a treatment and the demyelination stops and myelination/remyelination occurs.

Response: We have changed the title of this manuscript to: “Advantages of Adult Mouse Dorsal Root Ganglia Explant Culture in Investigating Myelination in an Inherited Neuropathic Mice Model”. Trembler-J (Tr-J) mice carry a spontaneous Leu-16-Pro (L16P) substitution in the first transmembrane domain of PMP22, which is identical to a mutation in humans with early-onset severe neuropathy (Suter et al., 1992). Nerves from heterozygous Tr-J mice show hypomyelination during early postnatal development, with pronounced demyelination and axonal atrophy mimicking the neuropathology of CMT1E and Dejerine-Scottas demyelinating disease (Notterpek et al., 1997). To further investigate the non-inherited neuropathic model, we plan to utilize this culture system as the reviewer suggested. 

Reference

  1. U Suter et al. (1992) Nature 356(6366):241-4
  2. L Notterpek et al. (1997) J Neuroscience 17(11) 4190-4200

2) The term ‘satellite Schwann cells’ (e.g. line 20, 40) used throughout the manuscript is incorrect as satellite cells are not Schwann cells. Furthermore, in lines 41-42, the authors wrote: ‘there is characteristically no difference between satellite Schwann cells of DRG and sciatic-derived Schwann cells’. This is incorrect as the paper (George et al., 2018) says: ‘similar rather than no difference’. Also, is it possible for the myelination to be due to the Schwann cells that have not been completely removed when the peripheral nerves were removed as the nerve fibers are present within the DRG?

Response: We thank the reviewer for pointing out this error, which has been corrected to ‘satellite glial cells’ and ‘Transcriptome analysis showed that satellite glial cells from DRG and sciatic nerve-derived Schwann cells were notably similar’ as the reviewer requested. We suspect the remained Schwann cells can form the myelin sheath, but mutated Schwann cells in an inherited neuropathic model may be not possible to make compact myelin.    

3) There are many grammatical errors throughout the manuscript making some parts confusing, so require editing by a native English-speaking scientist. For example, line 164: ‘declined become unstable’ and line 165 ‘not to be a stable period for forskolin stock’, and many others.

Response: We have done English Editing by a certified company Enago (www.enago.co.kr).

4) Missing the strength of the protocol which I assume is a simpler protocol with just one step rather than multiple steps. Maybe this should be in the abstract or introduction.

Response: We have edited some advantages of this protocol in the abstract as the reviewer suggested. 

Line 17. “However, 3D DRG explant neurons and myelination cells in Matrigel-coated culture are obtained from the same mouse, and the culture period is shorter than that of 2D co-culture, and fewer animals are needed. In addition, it has simpler and shorter experimental steps than 2D co-culture.

5) Removing the sheath surrounding the DRG is difficult, so what suggestion do the authors have as there are many methods?

Response: There is no special method. Unlike embryo DRG in the case of adult DRG, removing the sheath surrounding the DRG was necessary for neurite growth and cell migration, so it was carefully removed using forceps. We have used the following basic protocol of reference (M Fornaro et al., 2018).

Reference

  1. M Fornaro et al. (2018) Journal of Visualized Experiments 133, e56757

6) Line 170. How was ‘gently washed’ carried out? Pipette slowly? Any swirling or on a slow rocker? How was PBS removed between washes?

Response: We have changed to the detailed method as the reviewer commented.

Line 177 “~ it was gently washed in 1× PBS for 3 min using a pipette.

7) (Table 2) organisation unclear and need to be split up into antibody table and reagent preparation protocol table. Better transpose the table so similar to table 1 format and have the volume in a separate column for neatness. Also, avoid abbreviation GP and Rb.

Response: We apologize to the reviewer for the unclear information. We have changed and separated Table 2 and Table 3 as the reviewer suggested.

8) Figure 2.  Use a few arrows to show cell body if possible to help reader see cells. Upper scale bars should be black as unclear in white.

Response: We have edited Figure 2 using white arrows and arrowhead to indicate the node of Ranvier of myelin, and changed white to black scale bars. 

9) Figure 3. Unclear which part of the culture is adult or embryonic, and which is WT or Trembler J mouse? Also panel lettering ordering need to change (e.g. A and D should be A and B) for better flow in body text.

Response: We apologize to the reviewer for the unclear information. Figure 3 has been modified as the reviewer commented, and added with a quantitative analysis.  

10) Missing a discussion section to compare your methods with others.

Response: We have added the comparison with other methods in the discussion part.

Line 251. “The method of adult DRG explant culture for mechanism study and screening of the CMT1E model is described in this research. There are several differences compared with traditional co-culture. The main difference is that both the neurons and Schwann cells originate from one individual mouse. ~~~~

Minor concerns:

11) what’s the percentage of CO2 levels?

Response: We have edited it as “The CO2 level was maintained at 5% in all culture steps” in line 148.

12) Fig1. Last section. Should it be ‘neurite growth’ rather than ‘cell growth’?

Response: We have changed ‘cell growth’ to ‘neurite growth’ in figure 1.

13) Define ICC.

Response: We have changed all ICC to ‘immunocytochemistry’.

14) Line 115. Confused why 36h if only overnight is sufficient which is 24h.

Response: We have added detailed information to “If the gelation time exceeds 36 h, the Matrigel coating can be too dry to use.” in line 120. 

15) Line 150. ‘span’?

Response: We have changed ‘span’ to ‘duration’ in Line 156.

16) Table 1. Define ‘A/A’

Response: We have changed ‘A/A’ to ‘Antibiotic–Antimycotic’.

Reviewer 4 Report

The manuscript compares two different experimental models to study myelin sheet formation: an ex-vivo model and an in-vitro model. This paper is appropriate for Methods and Protocols and the results could address the use of the ex-vivo model with respect to the in-vitro one.

The manuscript is clearly presented with the details to reproduce the procedures. Moreover, the images are explicative and helpful.

Nevertheless, please consider the following suggestions/considerations:

- Animals: 

- if applicable, the authorization number of the Authority for the use of animals should be reported.

- the origin of the animals should be indicated. 

- The microscope used to obtain fig. 2-3 should be added.

- Why is the Result section not supported by statistical analysis? 

The discussion/conclusion section should be added as indicated in the “Instructions for Authors”; moreover, it is expected because in the abstract and introduction the final considerations have been shortly reported.

Author Response

Reviewer #4

The manuscript compares two different experimental models to study myelin sheet formation: an ex-vivo model and an in-vitro model. This paper is appropriate for Methods and Protocols and the results could address the use of the ex-vivo model with respect to the in-vitro one.

The manuscript is clearly presented with the details to reproduce the procedures. Moreover, the images are explicative and helpful.

Nevertheless, please consider the following suggestions/considerations:

Response: We thank the reviewer for these positive comments. We have edited the text as the reviewer suggested.

1) Animals: 

- if applicable, the authorization number of the Authority for the use of animals should be reported.

- the origin of the animals should be indicated. 

Response: We have added an animal authorization number from the Institute and the origin of trembler-J mice.

Line 281. “Institutional Review Board Statement: All animal procedures were performed in this study with the approval of the Institutional Animal Care and Use Committee of DGIST(DGIST-IACUC-20082701-0001, DGIST-IACUC-20082702-0002)

Line 60. “One CMT mouse model, Trembler-J (Tr-J) from Jackson Laboratory (strain no. 002504), involves spontaneous mutants carrying point mutations in the PMP22 gene

2) The microscope used to obtain fig. 2-3 should be added.

Response: We have added the information about the microscope to the text.

Line 107. “Axio observer Z1 (Carl Zeiss, Oberkochen, Germany; Cat. No. Observer.Z1)

3) Why is the Result section not supported by statistical analysis? 

Response: We have added a quantitative analysis result in Figure 3 as the reviewer suggested.

4) The discussion/conclusion section should be added as indicated in the “Instructions for Authors”; moreover, it is expected because in the abstract and introduction the final considerations have been shortly reported.

Response: The discussion section has been edited with more information as the reviewer commented.

 Line 250. “5. Discussion ~~

Round 2

Reviewer 3 Report

All the comments have been addressed adequately.